# The Influence of Hybrid Surface Modification on the Selected Properties of CP Titanium Grade II Manufactured by Selective Laser Melting

**DOI:** 10.3390/ma13122829

**Published:** 2020-06-24

**Authors:** Anna Woźniak, Marcin Adamiak, Grzegorz Chladek, Mirosław Bonek, Witold Walke, Oktawian Bialas

**Affiliations:** 1Faculty of Mechanical Engineering, Department of Materials Engineering and Biomaterials, Silesian University of Technology, Konarskiego 18A Street, 44-100 Gliwice, Poland; marcin.adamiak@polsl.pl (M.A.); grzegorz.chladek@polsl.pl (G.C.); miroslaw.bonek@polsl.pl (M.B.); oktawian.bialas@polsl.pl (O.B.); 2Faculty of Biomedical Engineering, Department of Biomaterials and Medical Devices Engineering, Silesian University of Technology, Ul. Roosevelta 40 Street, 41-800 Zabrze, Poland; witold.walke@polsl.pl

**Keywords:** SLM, Ti Grade II, corrosion test, PVD (Physical Vapour Depositions) coatings, laser texturing, wettability, EIS test

## Abstract

The human body is an extremely aggressive environment in terms of corrosion. Titanium and its alloys are one of the most popular biomaterials used for implant applications due to biocompatibility. However, every element introduced into the body is treated as a foreign body. The human body’s immune response may, therefore, lead to implant rejection and the need for reoperation. For this purpose, it seems important to carry out surface modifications by applying coatings and inter alia by texturing to implants. The objective of this paper is to investigate the effect of surface treatment on the chosen properties of the pure titanium (Grade II) samples obtained by selective laser melting (SLM) processing. The samples were divided into five groups: Initial state (after polishing), after surface modification by the physical vapour deposition (PVD) method—CrN and TiN coatings were deposited on the surface of the tested material, and after laser texturing. The paper presents the results of the microscopic investigation, chemical and phase compositions, and physicochemical and electrochemical properties of the tested samples. Based on the results obtained it can be concluded that the hybrid surface modification shows significant effects on the properties of the pure titanium. The samples with the textured PVD-deposited TiN coatings were characterized by favorable physicochemical properties and were the highest performing in terms of pitting corrosion resistance.

## 1. Introduction

Titanium and its alloys, especially pure titanium (Ti Grade II and Ti Grade IV) and Ti-6Al-4V alloy (Ti Grade V) are the most common metallic biomaterials for long-lasting implantation in dental and orthopedic application [1,2]. Increasing employment of titanium results from its fair biocompatibility, good corrosion resistance associated with the ability to spontaneously form a passive layer and high strength-to-density ratio [3,4,5,6,7,8,9]. However, the passive layer does not fully guarantee corrosion protection, because it can include some defects (inclusions and discontinuity—weak spots), which could become the initial areas of corrosion [10]. This is an important aspect, given that the human body environment is an aggressive corrosive environment. Additionally, the main disadvantage of titanium is poor wear resistance, which is generally inferior to that other metallic biomaterials [11]. It may lead to degradation of the surface thereby releasing and accumulating elements/corrosion products causing a metallosis [11]. Salem et al. [12] suggested that titanium implants components lead to more metallosis causes in comparison to cobalt-chrome implants. Therefore, modifications of the surface properties of the titanium implants are still recognized as an important way to improve widely understood biocompatibility [13,14]. Serval chemical and physical surface modifications, e.g., chemical etching, plasma treatment, ion implantation. physical vapour deposition (PVD) and atomic layer deposition (ALD) coating deposition methods are used [10,15,16,17,18,19]. Among the many different types of coatings currently used for biomedical purposes, very common and relatively easy to perform are TiN and CrN coatings, which are designed to increase not only the corrosion resistance of pure titanium but also to changing the wetting angle [19,20]. Surface chemistry (i.e., wetting angle) together with its surface morphology are critical factors that could affect osseointegration [21,22], so to increase the possibility of bone tissue growth, additional methods of treatments are considered, such as laser texturing [23,24,25,26,27]. Some work also suggests that surface texturing could be a method to enhance tribological properties [27].

Therefore, in the presented work we focused the surface modification of titanium Grade II manufactured by selective laser melting. The choice of this material was motivated by the fact that it is the main material used for dental implants [28,29,30] and it is an attractive basis for lowering the Young’s modulus [1,2,3,4,5,6,7]. Moreover, the dental implantology is also an area of potential application of using selective laser melting (SLM) technology [31,32]. The use of titanium allows also assessing the effect of treatment on the dominant component of its alloys, without an impact resulting from other components. To improve surface properties, we used the hybrid modification by the PVD method (CrN and TiN coatings deposition) and laser texturing. The aim of the presented work was to instigate the influence of hybrid coating on the chosen properties, such as the value of contact angle, surface free energy (SFE), corrosion resistance and wear resistance of the Grade II pure titanium samples manufacturing by selective laser melting. The selection of the investigated aspects is related with functioning of the implants. Wetting angle and surface free energy are important factors that could affect cellular response and bacterial plaque accumulation [21,22], surface topography (roughness, micro holes and scratches) is essential to promote osseointegration [22,33,34], while corrosion resistance and wear resistance are of great importance for the durability of implants [10,35]. Additionally, in order to investigate the degradation of the coating (obtained by the PVD method or spontaneously formed as a result of contact with oxygen or after texturing process) and impart more information about the corrosion mechanism the electrochemical impedance spectroscopy (EIS) test was used [36,37]. The EIS test provides information about the corrosion kinetics and electrochemical control mechanism [38].

## 2. Materials and Methods

The Grade II pure titanium powder (KAMB Company, Warsow, Poland) with chemical composition declared by the producer (Table 1), was used to prepare test samples. The powder material was spherical in shape, with size fraction in the range from 15 to 45 µm (Figure 1).

The tested samples were obtained by the selective laser melting process using the AM125 SLM machine (Renishaw Company, Gloucestershire, Germany). That system employs a continuous wave ytterbium fiber laser (YFL) with a wavelength of 1070 nm with the maximum average power P of 200 W, maximum laser scanning speed SP up to 2000 mm/s, and a laser beam diameter D equal to 35 µm. In order to minimize the process of oxidation and degradation of the used powder material, the processes were conducted under vacuum and with the protective atmosphere of high purity argon. For this reason, the working chamber was flooded with 99.996% pure argon until the value below 0.1% of oxygen was reached. The continuous flooding with argon guarantees a low oxygen content—oxygen content of around 100 ppm could be achieved. During the process, a pure titanium base plate was used. Before starting the process itself, the plate was subjected to polishing to guarantee good adhesion of the powder material. Additionally, the plate was heated up to 150 °C, and the temperature was kept at the same level throughout the whole process. The process technological parameters were optimized using a simplified model (Equation (1)), which specifies the density of energy input (Andrews number). According to the designation of the formula: P—laser power [W], SP—scanning speed [mm/s], t—layer thickness [µm], PD—point distance [µm] [39,40,41,42,43].
E = P/(SP × PD × t) [J/mm^3^](1)

The energy density delivered to the powder material was approximately 75 J/mm^3^ (Table 2). Samples selected for testing are characterized by a density above 95% of the theoretical density of pure titanium (4.51 g/cm^3^)—the density of the tested samples was measured according to the Archimedes buoyancy method, using type AS 220 R2 analytical balance (Radwag, Radom, Poland) with the measurement accuracy of ±0.1 mg. A meander scan strategy was employed. The cubic specimens with an edge length of 10 mm were built at 0° to the build direction. Process parameters, scan strategy, and samples’ orientation were designed with MARCAM AutoFab software (PresseBox, Baden-Württemberg, Germany).

In the next stage all tested samples were subjected to mechanical grinding, which was performed using a MD-Piano disc (MD-Piano 200, 600, 1200) in time t = 2 min per each gradient of disc and mechanical polishing, which was carried out with colloidal silica suspension OP-U 0.04 µm in time t = 10 min. The mechanical finishing process was performed using the grinding-polishing machine TERGAMIN-30 (Struers, Willich, Germany).

In the next stage the nitride layers were deposited on the surface of the tested samples by the PVD method. The specimens were divided into two groups, depending on the type of the coating—CrN and TiN. The CrN coatings were deposited by the sputtering method. The thickness of the obtained CrN layer was approximately ~2 µm. The TiN coatings were deposited by the arc evaporation method. The thickness of the obtained TiN layer was approximately ~4 µm. The specimens in the initial state were ultrasonically cleaned in a isopropyl alcohol for 15 min before surface modification.

Laser micro texturing of samples with PVD coatings was performed on selected samples. The experiment was accomplished by the A-355 Laser Micromachining system (Oxford Lasers, Didcot, UK) based on 355 nm wavelength diode-pumped solid-state picosecond laser. The system of pulsed laser beam emission was chosen due to the potential of achieving high energy density and the ability to perform ablation, i.e., the situation in which high-energy laser radiation quantum is able to decrease the strength of the bonds between the particles which, in effect, makes it possible to evaporate atoms layer by layer. The structure modification by a pulsed laser makes it easier for an extremely focused, pulsed light beam to fall to consecutive locations where the material is evaporated, resulting in the absorption of the energy of the photon by the activated atoms and their evaporation (ablation). The maximum pulse energy for the A-355 ps Laser Micromachining system is 0.2 mJ with an average power of 18 mW, the pulse duration is in the range of 5–10 ps. The control software system is Cimita (Oxford Lasers, Didcot, UK), which enables us to create, edit and run Computerized Numerical Control (CNC) programs. The texturing process parameters were presented in Table 3. The path of laser texturing was the tightly arranged regular hexagonal honeycomb pattern with the side length of 0.29 mm (Figure 2) [27,44,45].

Before testing, all samples were placed in an ultrasonic bath for 15 min. The ultrasonic cleaning process was performed with high purity ethyl alcohol. The indication of the tested samples is given in Table 4.

### 2.1. Surface Topography Investigation and Phase Analysis

In order to obtain higher magnification images of powder material and surface topography the scanning electron microscope Supra 35 (Zeiss, Oberkochen, Germany), equipped with a SE secondary electrons-type detector was used. The microscopic observations were conducted with an accelerating voltage of 20 kV.

In order to obtain additional information of surface topography, microscopic observations using an optical microscope were performed. The microscope VHX-7000 series (Keyence, Osaka, Japan) was used.

The surface topography analysis of the tested samples using atomic force microscopy was performed. The observation was conducted by the contact mode with the use of XE-100 atomic force microscope (AFM, Park System, Mannheim Germany). For the samples M1–M3 the surface roughness measurements were also performed. The characteristic parameters describing the surface roughness—a rough mean square, (RMS/R^q^), the arithmetic average of ordinates profile (R_a_) and the sum of maximum height and maximum depth (ΔZ) were calculated over 25 × 25 μm area scan. The calculations were carried out using a PC computer with XEI software (Park System). For each tested sample, 20 measurements were taken.

Phase analysis studies were performed using an X’Pert PRO X-ray diffractometer (Panalytical, Almelo The Netherlands) equipped with a cobalt lamp X-ray source. The lamp was set to 40 kV and the heater current of 30 mA was used. The X-ray phase analysis was carried out according to the Bragg-Brentano geometry, using PIXcel3D detector. Measurements were conducted within the 2Θ Bragg angle ranging from 20 to 90° with a step of 0.05 [46,47].

### 2.2. Physicochemical and Electrochemical Properties Analysis

Sessile drop contact angle measurements were performed on a Surftens Universal Goniometer (OEG, Frankfurt Germany) equipped with a camera for taking photos of drops of measured liquids and a PC computer with Surftens 4.5 software to analyze the recorded drop image. Distilled water (POCH S.A., Gliwice Poland) and diiodonomethane (Merck, Warsow, Poland) were used as measurement liquids, with a drop of 1 µL in volume placed on the surface of the samples. After 20 s from when the drop was dripped the measurements were taken and the duration of one measurement was t = 60 s. The study was carried out at room temperature T = 23 ± 1 °C (289 K). For all tested samples five measurements using both measurement liquids were performed, and the average value was determined. Based on the obtained values of the contact angle, the Surface Free Energy (SFE) calculation according the Owens-Wendt method were taken. The values of SFE and their dispersion and polar components were given in Table 5 [43,48,49].

In order to determine the pitting corrosion resistance, a potentiodynamic test by recording the anodic polarization curves was performed. The pitting corrosion test was carried out according to the PN–EN ISO 10993-15 standard [50,51]. The measurements were carried out with the use of test stand comprised of an Atlas 0531 EU potentiostat (ATLAS-SOLLICH, Rębiechowo Poland), PC with AtlasLab software to saving recorded polarization curves and electrochemical cell with three-electrode system. The working electrodes were represented by the tested sample (anode), platinum wire (PtP-201) was used as an auxiliary electrode, and the reference electrode was a saturated Ag/AgCl electrode. The corrosive tests started with the establishment of the open circuit potential E_OCP_ at electrodes condition. Recording of the polarization curves was started from the value of the initial potential E_init_ determined according the formula E_init_ = E_cop_ − 100 mV. The potential value changed along the anodic direction at 1 mV/s rate. Once the anodic current density i = 1 mA/cm^2^ or the maximum measuring range reached + 4000 mV, the polarization direction was changed. On the basis on the recorded curves, the characteristic parameters describing pitting corrosion resistance were determined—The corrosion resistance E_corr_ [mV], breakdown potential E_b_ [mV], transpassivation potential E_tr_ [mV] and repassivation potential E_cp_ [mV]. The impedance measurements were taken with the use of AutoLab’s PGSTAT 302N system (AutoLab, Warsow Poland), equipped with the frequency response analyser (FRA2) module. The tests were carried out using a three-electrode system—Identical to that used in a potentiodynamic test. This measurement system made it possible to perform tests within a 10^4^–10^−3^ Hz frequency range. The voltage amplitude of sinusoidal signal activating amounted to 10 mV. Based on the performed tests, impedance spectra of the system were determined (Bode and Nyquist diagram) and the measurement data were compared against the equivalent system. The obtained measurement data were adjusted through the method of the smallest squares to the substitute setup and the values of resistance R and capacity C were determined. The testing environment was identical as in the pitting corrosion test. All electrochemical tests were carried out in the Ringer solution (NaCl—8.6 g/cm^3^, KCl—0.3 g/cm^3^, CaCl_2_ 2 H_2_O—0.33 g/cm^3^) at the temperature T = 37 ± 1 °C and pH 6.9 ± 0.2.

The microscopic observation of the surface of the tested samples before and after potentiodynamic tests was performed using an Axio Observer Z1 Microscope (Zeiss, Oberkochen, Germany).

### 2.3. Wear Test

The wear test was performed using the pin-on-disc method. The test was carried out using a CSM tribometer (CSM Instruments, Needham, MA USA), equipped with an arm with a pin holder, controller and the rotated stage. A ceramic ball Al_2_O_3_ with the 6 mm and 1300 HV hardness dimeter was used as a counter specimen. The test was carried out with the load of 10 N and the linear speed was 10 cm/s. Prior to wear tests, both the tested samples and ball were ultrasonically cleaned in acetone. Based on the performed measurements, the value of the coefficient of friction µ was determined. All the tests were performed in ambient air conditions with temperature in the range of 18–22 °C and humidity an approximately 52–60%. For each samples group, to guarantee the stability and accuracy of the experimental data, each test was repeated five times. The wear volume was calculated based on the cross-sectional area of the wear tracks, that were measured using a profilometer (Surtronic, Taylor Hobson, UK). The wear factor W was calculated according to Equation (2) [52], were V is wear volume [m^3^], F_n_—force [N], S—total sliding distance [m].
W = V/F_n_ × S(2)

## 3. Results

### 3.1. Surface Topography and Phase Analysis

The samples in the initial state M1 were fabricated by the SLM method successfully. The surface of the tested samples subjected to the surface modification were homogenous and no unmelted powder particles and porosity or other defects could be seen. The mean value of the R_a_ parameter for the M1 samples group, obtained by the AFM method for 25 × 25 µm was an approximately 84 ± 14 nm (Table 6). The X-ray diffraction (XRD) patterns (Figure 3c) of samples in initial state M1 resulted in only alpha, which confirms the data contained in the material card, which was provided by the manufacturer. The scanning electron microscopy (SEM) and AFM (Figure 3) results of samples after PVD surface modification are continuous, no delamination in coating continuity was observed. Nitride layers were uniform, with no spalling and discoloration, surface was relatively smooth because the subtle scratches on the polished surface (Figure 3a) were covered by the CrN or TiN films (Figure 3d,g) [53]. However, there are a lot of microparticles on the surface of TiN coating, which is a common phenomenon in the arc evaporation PVD process. Additionally, it is noted that CrN coating, deposited by sputtering method exhibit more uniform surface morphology, without the formation of particles in the coatings. The particles on the surface of the M2 and M3 sample visible in Figure 3 can be Cr and Ti agglomerates less frequently observed for CrN. Based on the obtained results changes in morphology of the coatings deposited can be observed, resulting in elevated surface roughness. The highest values of surface roughness parameters were obtained for the samples with CrN layer M2 and the mean value was 187 ± 24 nm. The deposited TiN layer (M3 samples group) proved to decrease the surface topography compared to the samples in the initial state Grade_is. The lowest value of surface roughness was obtained for the Grade_TiN sample group and the mean value of R_a_ parameter was 42 ± 9 nm (Table 6). Based on the local chemical analysis it can be concluded, that the process of the modification was properly carried out, according to the assumptions. The XRD spectrum shows that CrN coating and the TiN coatings (Figure 3f,i) have strong (113) and (111) preferential orientation, respectively.

The same textures were generated with the same laser processing parameters on both samples’ groups after surface modification (Figure 4). The diameter of the dimples was very even and fitted the design well. It can be seen from Figure 4d that the dimensions of the generated honeycomb were consistent with the CAD’s model dimensions. The dimension of the circle inscribed for both sample groups after texturing a single hexagon was approximately D = 0.475 mm, which was comparable to the dimensions of the CAD model, where the diameter was D = 0.5 mm. Based on the microscopic investigation it can be found that grooves present typical laser induced surface structures, which were formed by the interaction of laser beam and sample surface. The textured surfaces obtained by laser ablation have been characterized as continuous groove, characterized by a non-regular explosively evaporated bottom (Figure 4b). Corrugated grooves with holes and chaotic structures especially for M4 samples group have also been observed (Figure 4c). For the M4 and M5 sample groups, this formed noticeable areas with slightly different colors, which is the effect of various oxide films that were formed on the surface of the treated areas. The texturing process was carried out without a protective atmosphere.

In the next stage the depths of the grooves of the hexagon were measured—The groove depth was measured in detail. The groove depth for the samples with CrN layer (M4) was approximately 1.5 ± 0.21 µm (Figure 5). The measurements were conducted using a digital microscope where it was found that the highest values of groove depth were recorded for the samples with TiN layer (M5) and the mean value was 4.5 ± 0.15 µm (Figure 6). Based on the measurements of the depth of generated textures, no thicknesses of previously refined CrN layers were exceeded, and thickness of the CrN layer was ~2 µm. For a TiN layer with a thickness of approximately ~4–5 µm, the generated texture may slightly interrupt the continuity of the layer.

### 3.2. Physicochemical and Electrochemical Properties Analysis

The results of wettability measurements and surface free energy calculated for all tested samples were presented in Table 7. Test examples of drops dripped on the surface of the samples are presented in Figure 7. The results obtained pointed to the increased value of wetting angle θ for the samples with PVD deposited layer and decreased for the samples after texturing the samples in initial state (M1) and the samples with CrN and TiN layer after texturing (M4, M5) were hydrophilic and showed the relative good surface wettability, with mean values for three sample groups being θ = 78 ± 10°, θ = 79 ± 11°, θ = 85 ± 11°, respectively. The surface modification by the PVD method effected in the elevated wetting, regardless of the type of the layer, resulting therefore in the hydrophobic surface. The highest value of contact angle was obtained for the samples with TiN layer (M3) and the mean value was approximately θ = 101 ± 12°. The wetting angle values for these sample groups obtained were θ_av_ > 90°. The differences were observed in the values of the surface free energy SFE throughout the investigated sample surfaces. However, values of the contact angle, using diiodonomethane as a measure liquid, were in the range from 44 ± 6°for the M5 samples group to 51 ± 9° for the M1 samples group. The values of the SFE for all tested samples were in the range of approximately 23 mJ/m^2^ for the M3 sample group to 44 mJ/m^2^ for the M1 sample groups. Additionally, the highest value of apolar components was higher compared to polar components. Based on this it can be concluded that all tested samples exhibited a greater affinity to apolar groups of SFE than to polar ones.

The purpose of potentiodynamic testing was to evaluate the pitting resistance in form of anodic polarization curves shown in Figure 8. Additionally, the characteristic values describing the pitting corrosion resistance are presented in Table 8. Only for the samples in initial state M1 were the hysteresis loops recorded; the existence breakthrough potential E_b_ and the repassivation potential E_cp_, proving initiation and development of pitting corrosion, were measured (Figure 8). The mean values of breakdown potential and repassivation potential were E_b_ = 1379 ± 87 mV and E_cp_ = 1379 ± 87 mV, respectively. For other sample groups the progress of anodic polarization curves was characteristic for the samples with high pitting corrosion resistance. The existence of the transpassivation potential has been stated. The highest value of transpassivation potential was observed for the 5th sample group M5 (the samples with TiN layer and after texturing treatment) and the mean value was approximately E_tr_ = 3052 ± 143 mV. Similar results were observed for the samples with that same layer M3 and the average value of transpassivation potential was E_tr_ = 2572 ± 102 mV. However, in samples with CrN film deposited by the PVD method, lower values of transpassivation potential were noted, regardless of the presence of the texture on the surface. The values for M2 and M4 were similar and were in the range from 1088 to 1104 mV. Additionally, the values of transpassivation potential for both sample groups with CrN layer were lower comparable to the value of breakdown potential established for the samples in initial state M1, which indicates lower resistance to pitting corrosion.

During the next stage, EIS tests were carried out to evaluate the electrochemical properties and obtain additional information about corrosion resistance. The results in the form of impedance spectra for all tested samples are presented in Figure 9. The determined impedance spectra point to diversified kinetics of the corrosion processes taking place in the system-tested samples, using Ringer solution. The Nyquist diagrams for all tested samples presented fragments of semi-circles, which is typical for the thin layer (Figure 9a,c,e,g). The presented semi-circles were deformed to a different degree, in some cases from the linear relation between the components of the imaginary impedance Z” and the real one Z’. For the samples with TiN layer and after texturing the fragments of semi-circles were placed near the beginning of the coordinate system and the biggest were observed, which pointed to better corrosion resistance compared to the other sample groups. The maximum value of phase displacement at a broad range of frequencies presented in Bode diagram (Figure 9b,d,f,h) for the samples in initial state M1 was approximately Θ = 70°. Additionally, the values of the maximum value phase displacement for the samples obtained with the same layer (before and after texturing) were similar and were Θ = 50° for the samples with CrN (Figure 9d,g) coat and were in the range Θ = 65–70° for the samples with TiN coat (Figure 9e,h). For all sample variants, the inclinations log|Z| at the whole scope of frequency change are close to −1 which indicates the capacity character of the porous layer.

The analysis of the impedance spectra of corrosive systems tested samples using Ringer’s solution, the equivalent electrical system shown in Figure 10 was used. Based on this, the parameters of the electrical components of the replacement circuits describing the corrosion systems were determined—See Table 9. This method allowed the analysis and interpretation of processes and phenomena occurring at the interface: Tested samples - Ringer’s solution. Impedance spectra (Figure 10) obtained for the analyzed samples were interpreted by comparison to a substitute electrical system, which indicates the occurrence of two sublayers: A compact internal and a porous external layer (two time constants visible on the graph), where R_s_ is the resistance of electrolyte (Ringer solution), R_pore_—electrolyte resistance in the porous phase, CPE_pore_—capacity of the double layer (porous, surface), R_ct_—resistance of the deposited coating (characterizes the electric charge transfer resistance at the boundary of phases: Grade II—surface layer—Ringer solution) and CPE_dl_—capacity of the deposited coating, Figure 10. The use of two constant-phase elements in an electrical substitute circuit positively influenced the quality of fitting the experimentally determined curves. The mathematical impedance model of the above system is also presented in the Equation (3).

(3)$$Z=R_{s}+\frac{1}{\frac{1}{R_{p}}+Y_{01}{\left(j\omega \right)}^{n_{1}}}+\frac{1}{\frac{1}{R_{\mathrm{ct}}}+Y_{02}{\left(j\omega \right)}^{n_{2}}}$$

The fifth sample group M5 was characterized by high electrochemical stability, which was proved by relatively high value charge transfer resistance R_ct_. The mean value of R_ct_ parameters for M5 was 11,320 kΩcm^2^, which was similar to the value obtained for the M3 sample group, 10,254 kΩcm^2^.

The microscopic observations were also performed for all tested samples after the pitting corrosion test. Only for the samples with CrN coating M2 was the pitting corrosion observed (Figure 11). This means that the coating may not be a good protection against the environment.

### 3.3. Wear Test

Based on the obtained results of wear test (Table 10) it can be concluded that the samples in initial state M1 and samples with CrN layer M2 were characterized by the lowest value of friction coefficient, and the mean value was approximately µ = 0.54 ± 0.06 and µ = 0.55 ± 0.08, respectively. The laser texturing for both PVD layer variants improved to the increased value of the friction coefficient. The highest values were measured for the samples with the TiN layer after surface texturing M5 and the mean value was 0.69 ± 0.07.

Based on the obtained time history of the friction coefficient for the M5 sample group, there is a running a sudden increase in the coefficient of friction is initially visible, after which the friction coefficient has reached a steady state (Figure 12a). Based on the obtained results of the measurements of the depth of wear, for both CrN and TiN coating as a result of friction, the coating was damaged and permanently removed from the surface of the tested samples at the place of contact of the rubbing pair of the tested surface of the counter-sample. However, in the case of sample M5, the TiN coating continued to break after 500 s, where for the CrN coating the break occurred after about 125 s—That is according to the assumption since the thickness of the coatings are different. Based on the wear profile measurements the wear volumes were calculated. Based on the obtained results it can be concluded, that the surface modification, regardless of the type of the layer and laser texturing leads to reduced values of wear volume. Additionally, based on the wear coefficient calculated it can be concluded that the lowest volume was obtained for M5 sample group and the mean value was approximately W = 7.2 × 10^−17^. Examples of the results of the wear test for the M5 sample are presented in Figure 12b.

## 4. Discussion

For medical application, very important are the high quality of the surface of the materials and homogeneous structure. The surface microscopic observations of the samples in initial state (M1 samples group) are characterized by defect-free homogeneous structure. The pattern for the base material was typical of the pure titanium, and predominantly showed peaks associated with the α-Ti phase (hexagonal cell), which is comparable to other results presented by Stwora et al. [42]. The surface of the samples after modification by PVD, independently of the type of the nitride layer was uniform and no spalling or discolorations were detected. During modification, agglomerations formed on the material surface and the surface quality of the degraded coating, which was also stated in the work of Tang et al. [54]. The XRD analysis, showed a face centered cubic cell of CrN pattern with (113) and (200) peaks for the M2 samples—similar results were obtained by Chen et al. [55]. The diffraction pattern show also reflections from body centered cubic Cr. The pattern for the M3 sample group is predominantly that of the face centered cubic cell of TiN phase and the strongest peak is located at (111) crystal plane, which is comparable to the results obtained by Nolan et al. [56]. As (111) crystal orientation represents a compact crystal plane, it indicates that the coating structure obtained at this temperature is denser. In the case of ablation with an ultrafast pico-second laser, the texturing material absorbs heat, which causes melt or evaporation. During the interaction between laser and material, the molten and evaporated parts solidify, resulting in sputtering accumulation on the unprocessed surface [57]. The microscopic observation of the laser surface texturing of the samples after modification by the PVD method showed a coarse ablated surface, where the edge of the groove appears to be continuous; only a few corrugated grooves with chunks of residues scattered inside the groove and the heat-affected zone on the treated surfaces have also been observed. In the position close to the edge of the hexagon there is a little accumulation of the molten matters, which leads to a convex structure [27,58,59].

The micro- and nanoscale features of topography influence cellular biological interaction and interfere with macromolecules, which form conditioning films under human body physiological conditions. Ting et al. [60] has shown that the value of wetting angle of the pure titanium regardless of surface roughness was in the range from 70 to 90°, which is comparable to the obtained value for the M1 samples group (θ = 78 ± 10°). Based on the results obtained, it can be concluded that surface modification by the PVD method lead to increased wettability of the surface. The higher values of the contact were obtained for the samples with CrN layer (M2) and TiN (M3) and the mean value was θ = 98 ± 8° and θ = 101 ± 12°, respectively. For this case, the surface tension force leads to a minimization of the total surface energy of measuring the liquid drop in the effect of producing a spherical shape with a relatively high value of wetting angle. The value of the contact angle for the samples after laser texturing (M4 and M5) decreased, which can be associated with development of the surface topography and chemical heterogeneity, thus inducing deviations from an ideal surface. Due to increased surface area, there was an increased wettability of the textured surface against deionized water, and similar results were also obtained by Pfleging et al. [61]. After laser texturing, in the effect of the condensation of the molten material, some raised frames around the scanning path were formed. Based on the results obtained, it can be concluded that the formed frames cannot support the droplet of the water, and the droplet immersed into the texture, which is associated with the Wenzel state [44]. However, the values of wetting angle for both M4 and M5 are approximately 80°, which indicates a slight advantage of gravitational forces over surface tension forces. The different tendency of wetting angle between the distilled water and the diiodonomethane has been observed. The differences of the values are associated with their chemical character; the water is a polar and diiodonomethane is an apolar liquid, which differ in their vapour pressure at room temperature and the molecular volume [62]. On the weakly polar surface of the material, the lower values of the contact angles were measured for diiodomethane droplets—for all tested samples groups, the value of the contact angles measured using diiodomethane were similar and were in the range 44–51° and higher compared to polar components. In recent years, it has been proven that the degree of microbial colonization depends not only on the surface quality of the implant, but also on the hydrophobicity of the microorganisms, which was pointed to by Loosdrecht et al. [63]. Based on the results presented by Kochkodan et al. [64] and Giaouris et al. [65], it can be concluded the more hydrophilic cells or bacterial adhere more strongly to hydrophilic surfaces, while hydrophobic microorganisms strongly adhere to hydrophobic surface. Additionally, Borecká-Melkusová and Bujdakova et al. [66], showed that the microorganisms in the effect of changes in environmental composition, such as temperature or compositions of nutrients, can switch between hydrophilic and hydrophobic phenotypes. For example, the main driving force for adhesion of pathogens *S. Aurerus* and *S. epidermidis* is hydrophobicity. In addition, it was recorded that surfaces become more hydrophilic with higher values of the polar component of SFE [64] which was also obtained in this work for the surface of the tested samples. The samples with the highest value of the contact angle, exhibit a strongly affinity to apolar components of SFE. The SFE is a measure for the adhesiveness of the surface. For the material in initial state M1, the higher value of the SFE was recorded (γ_S_ = 44 mJ/m^2^). Apart from microbiological properties, the higher value of SFE of substrate material can affect the greater strength of PVD layers adhesion to the substrate. Additionally, the high value of SFE of the coatings can also affect the increase of wear resistance. It is important, because the main limitation of the use of titanium in human body is the aggressive environment present in vivo as well as low tribological properties and low resistance to the pitting corrosion. Based on the results obtained it can be concluded that, the laser texturing lead to improvement of corrosion resistance and it is possibly due presence of oxide. A detailed analysis of polarization behavior shows that the transpassivation potential increases from 1088 ± 87 mV to 1104 ± 95 for the M4 samples group, and from 2572 ± 102 mV to 3052 ± 143 mV for the M5 samples group. Similarly, the improvements in corrosion resistance of excimer laser surface treated Ti-6Al4V alloy was also reported by Yue et al. [67] and Kumari et al. [68]. The oxide layers, formed as a result of laser texturing in contact with corrosive environments, were characterized by greater stability and provides better protection against surface digestion. In the aspect of the hybrid surface modification and their impact on the physicochemical properties, the results obtained for the pure titanium and for the Ti-6Al-4V alloy may be similar. For both materials, the most commonly identified layer is TiO_2_, showing two different crystalline phases, anatase and rutile. For the Grade V, the passive layer may consist with a small amount of Al_2_O_3_ oxides [69]. Only for the samples in initial state, the existence of the breakdown potential has been stated. However, the microscopic observation didn’t showed the pitting. Additionally, surface corrosion after the potentiodynamic test was visible for samples with CrN coating (M2 samples group). It should be noted that the mean value of the breakdown potential for M1 samples group (E_b_ = 1709 ± 87 mV) was higher in comparison to the mean value of the transpassivation potential of M2 samples group (E_tr_ = 1088 ± 87 mV), which showed that the reconstruction of the passive layer on the surface of the M2 samples group began earlier than changes in the surface of M1 samples. Additionally, that the samples with the higher value of the contact angle and low value of the polar component of the SFE were characterized by higher corrosion resistance and better wear resistance—the samples with TiN layer after laser texturing M5 are characterized by the best pitting corrosion resistance—and the highest value of transpassivation potential E_tr_ = 3052 ± 143 mV and the highest value of the charge transfer resistance R_ct_ = 1132 kΩcm^2^ were obtained. For all tested samples, the EIS test indicates the occurrence of two sublayers: a compact internal and a porous external layer. The double layer is a good protection against a corrosive environment. The value of the resistance of charge transfer across the phase boundary—passive layer—of the Ringer’s rosettes informs about the amount of ions released from the surface of the tested material.

Additionally, the M5 sample group was characterized by the highest wear resistance. In case of the samples with CrN layer, texturing improved the decreased physicochemical properties, which can be associated with the oxide layers formed on the surface during laser investigation.

## 5. Conclusions

The value of the contact angle increases as an effect of the surface modification by the PVD method. However, for the samples after laser texturing, decrease of the wetting angle, as a result of surface development, were recorded. The higher value of the contact angle and surface free energy lead to better corrosion and wear resistance. As a result, the balance between the required biological reaction, elimination of bacterial adhesion and increase of corrosion and wear resistance needs to be considered. The samples with TiN layer after texturing M5 were characterized by the most favorable properties—a high value of the contact angle (near to hydrophobic character of the surface), low surface development, the best pitting corrosion resistance, and the highest value of the friction coefficient. In the future, in vitro biological studies should be carried out to confirm the relationship between the chemical character of the surface treatment and cellular response and bacterial adhesion.

## Figures and Tables

**Figure 1 materials-13-02829-f001:**
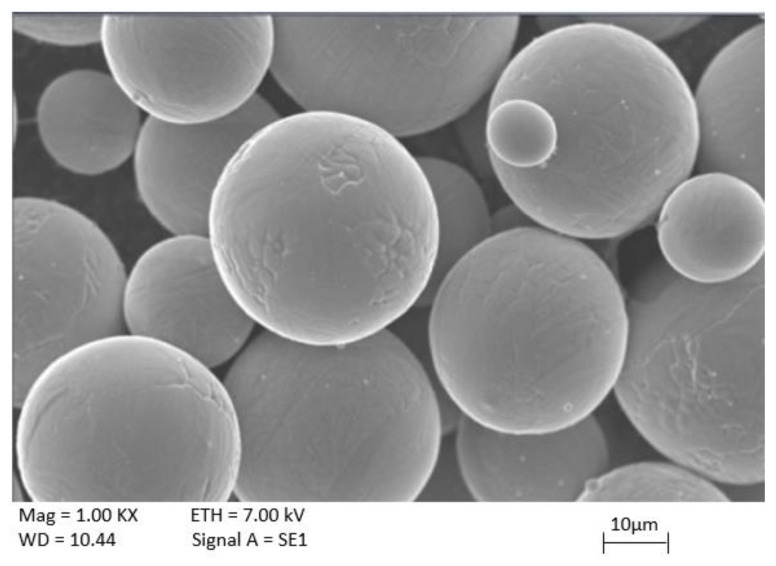
Scanning electron microscope (SEM) image of CP Ti Grade II powder morphology.

**Figure 2 materials-13-02829-f002:**
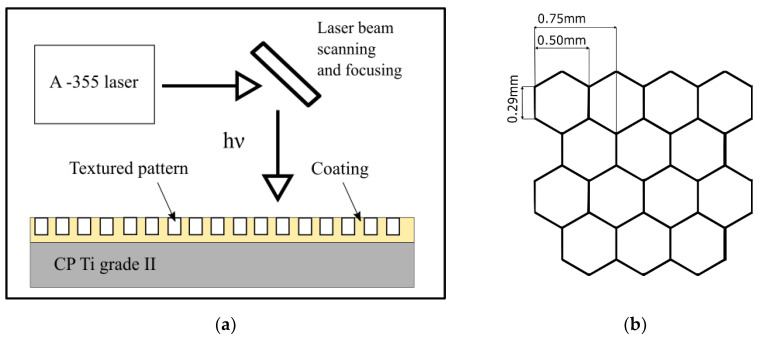
Schematic diagram of (**a**) laser micromachining system and (**b**) the honeycomb-like pattern of laser texturing.

**Figure 3 materials-13-02829-f003:**
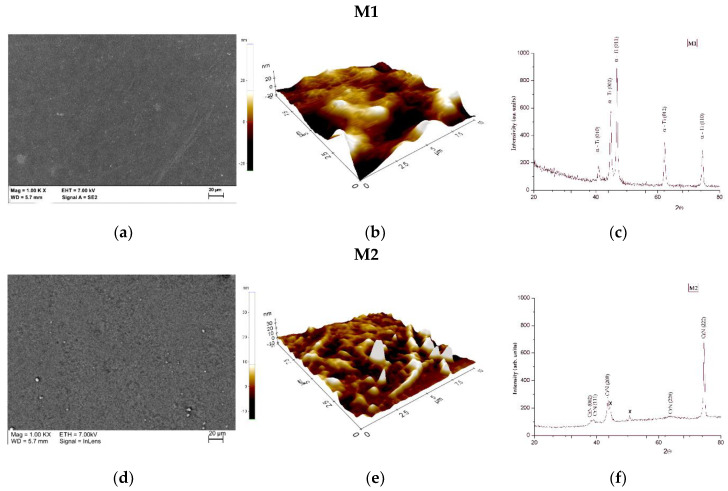
Images of surface topography phase composition measurements, for M1 samples group (**a)** SEM image; (**b**) atomic force microscopy (AFM) results (**c**) X-ray diffraction (XRD) pattern; for M2 samples group (**d**) SEM image; (**e**) AFM results (**f**) XRD pattern and for M3 samples group (**g**) SEM image; (**h**) AFM results (**i**) XRD pattern.

**Figure 4 materials-13-02829-f004:**
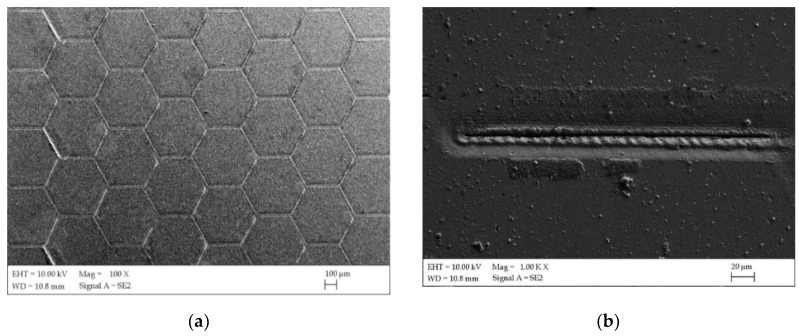
Examples of surface topography of the samples after texturing (**a**) SEM image M4; (**b**) SEM image single track M4; (**c**) SEM image single track M4; (**d**) optical microscopy M4.

**Figure 5 materials-13-02829-f005:**
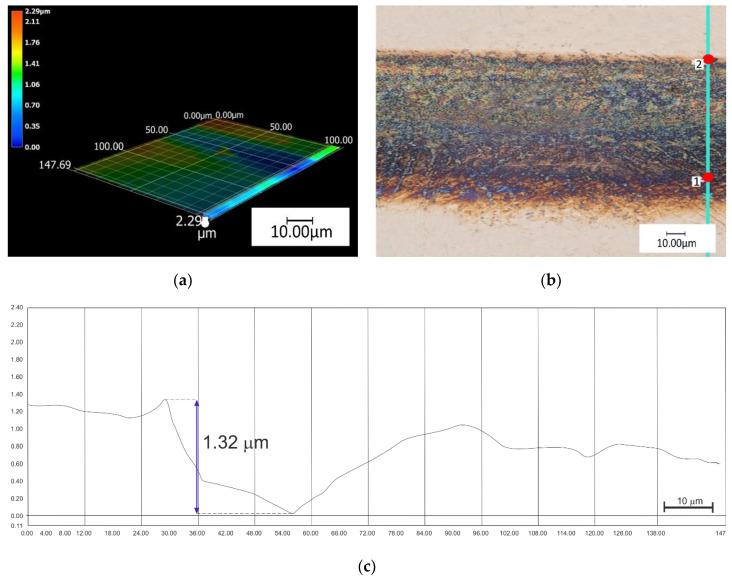
The results of digital microscopy observation for the 4th sample group M4: (**a**) 3D visualization of a single path; (**b**) single track with an indication of the depth measurement location; (**c**) groove depth measurements.

**Figure 6 materials-13-02829-f006:**
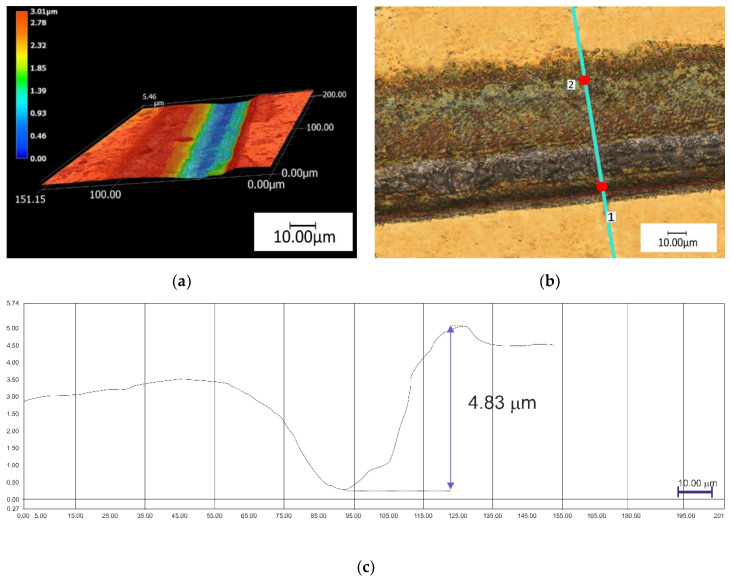
The results of digital microscopy observation for the sample group M5: (**a**) 3D visualization of a single path; (**b**) single track with an indication of the depth measurement location; (**c**) groove depth measurements.

**Figure 7 materials-13-02829-f007:**
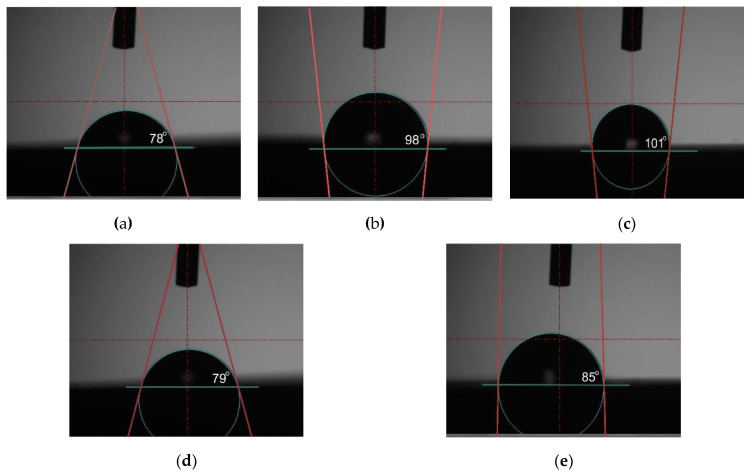
Sample measurement of wetting angle (**a**) M1; (**b**) M2; (**c**) M3; (**d**) M4; (**e**) M5.

**Figure 8 materials-13-02829-f008:**
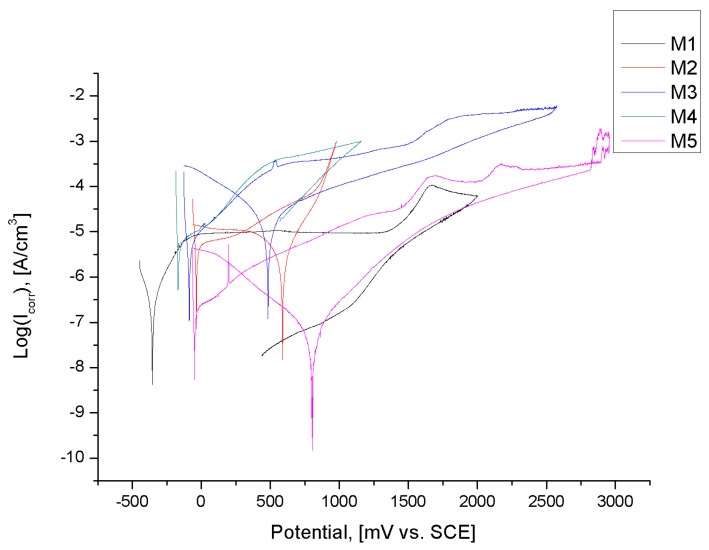
Example polarization curves.

**Figure 9 materials-13-02829-f009:**
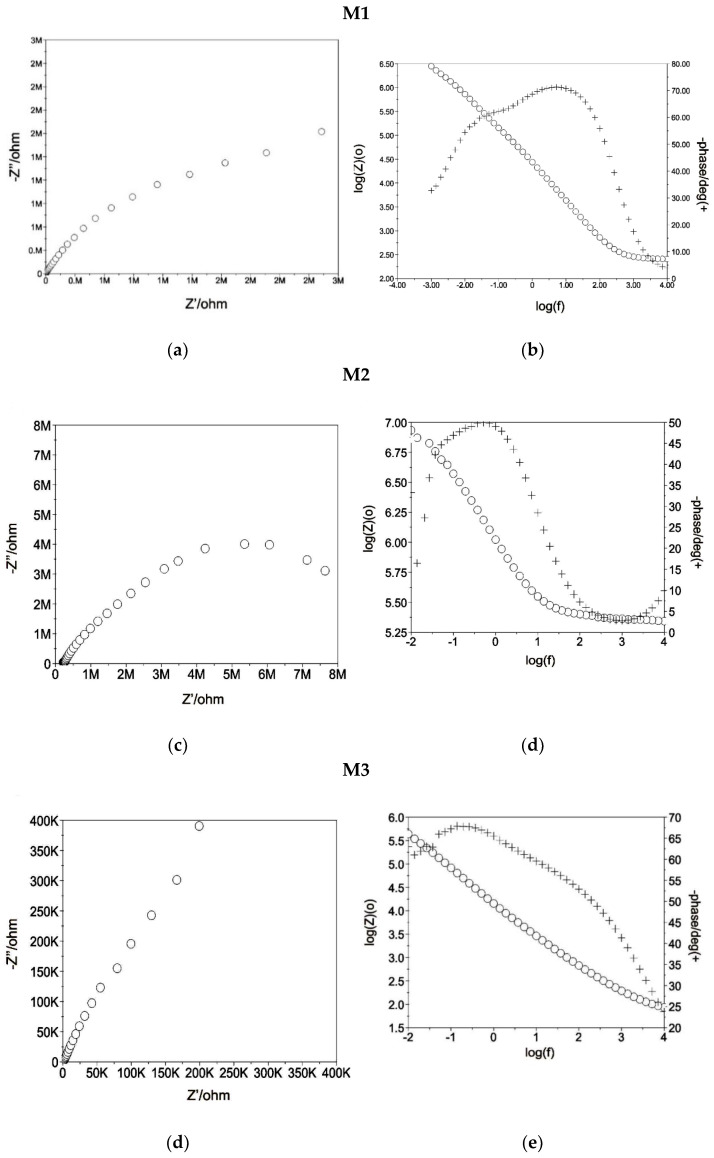
Examples of results of electrochemical impedance spectroscopy (EIS) test for M1 (**a**,**c**,**e**,**g**) Nyquist diagram; (**b**,**d**,**f**,**h**) Bode diagram.

**Figure 10 materials-13-02829-f010:**
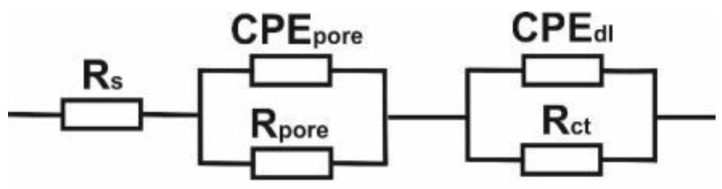
Electric substitute scheme.

**Figure 11 materials-13-02829-f011:**
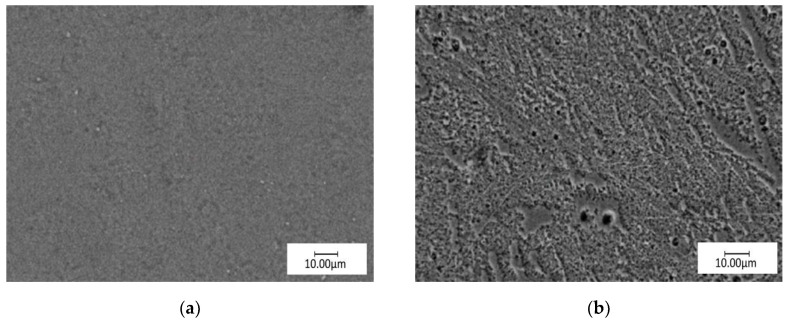
Examples of surface topography of M2 sample group (**a**) before; (**b**) after pitting corrosion test.

**Figure 12 materials-13-02829-f012:**
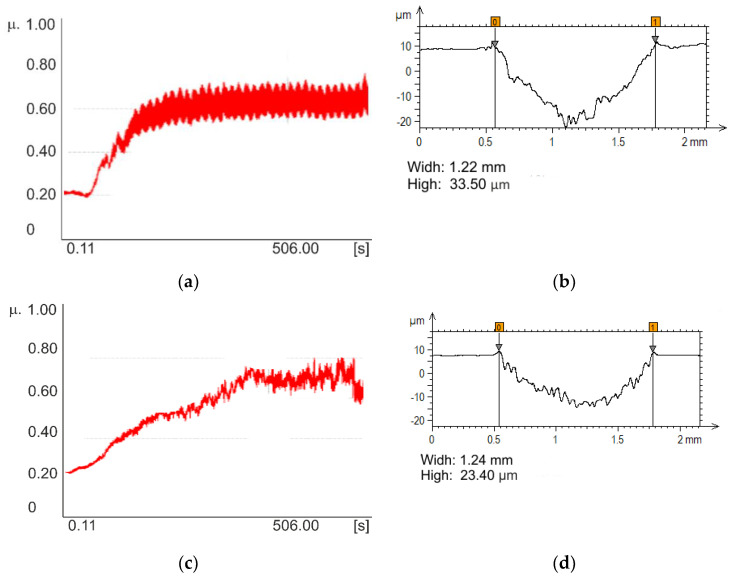
Examples of results of wear test for the M4 samples group: (**a**) diagram of coefficient friction changes; (**b**) wipe profile, and results of wear test for M5 samples (**c**) diagram of coefficient friction changes; (**d**) wipe profile.

**Table 1 materials-13-02829-t001:** Chemical composition of pure titanium Grade II powder used in experiment.

Ti	Fe	O	C	N	H
99.70	0.03	0.11	0.01	≤0.01	0.02

**Table 2 materials-13-02829-t002:** Process parameters and obtained density value of test samples.

Parameters	Value	Density of the Tested Samples ρ, [g/cm^3^]
Laser power P, [W]	200	4.31 ± 0.02
Scanning speed SP, [mm/s]	1333	
Layer thickness t, [µm]	25	
Poind Distance PD, [µm]	80	
Energy Density E, [J/mm^3^]	75	

**Table 3 materials-13-02829-t003:** Laser texturing process parameters.

Cutting Speed	Frequency	Laser Power at Laser Head	Etching Speed	Beam Width	M^2^
1 mm/s	400 Hz	48 mW	1 mm/s	~30 µm	<1.2

**Table 4 materials-13-02829-t004:** Identification of the tested samples.

No.	Name	Surface Treatment
1	M1	Initial state
2	M2	Initial state + CrN
2	M3	Initial state + TiN
4	M4	Initial state + CrN + texturing
5	M5	Initial state + TiN + texturing

**Table 5 materials-13-02829-t005:** The values of surface free energy (SFE) and their polar and apolar components for measuring liquids used in the Owens–Wendt method [43,48,49].

	SFE, [mJ/m^2^]		
	γ_L_	$\gamma_{L}^{d}$	$\gamma_{L}^{P}$
Distilled water	72.80	21.80	51.00
Diiodonomethane	50.80	50.80	0

**Table 6 materials-13-02829-t006:** Results of AFM surface roughness measurements.

No.	Name	Area, [µm]	R_a_, [nm]	R_q_ (RMS), [nm]	∆Z_max_, [nm]
1	M1	25 × 25	84 ± 14	98 ± 16	377 ± 98
2	M2	25 × 25	187 ± 24	220 ± 31	966 ± 105
3	M3	25 × 25	42 ± 9	82 ± 10	162 ± 20

**Table 7 materials-13-02829-t007:** Results of contact angle measurements and SFE calculated.

No.	Name	Wetting Angle, [°]		Surface Free Energy, [mJ/m^2^]		
		Distilled Water	Diiodonomethane	γ_S_	$\gamma_{d}^{S}$	$\gamma_{p}^{S}$
1	M1	78 ± 10	51 ± 9	44	25	10
2	M2	98 ± 8	47 ± 10	31	21	5
3	M3	101 ± 12	47 ± 10	33	17	4
4	M4	79 ± 7	45 ± 7	37	36	8
5	M5	85 ± 11	44 ± 6	37	32	6

**Table 8 materials-13-02829-t008:** Results of pitting corrosion test—mean values and standard deviations.

No.	Name	E_corr_, [mV]	E_b_, [mV]	E_cp_, [mV]	E_tr_, [mV]
1	M1	−355 ± 52	1709 ± 87	1379 ± 87	-
2	M2	−258 ± 21	-	-	1088 ± 87
3	M3	−91 ± 14	-	-	2572 ± 102
4	M4	−57 ± 11	-	-	1104 ± 95
5	M5	−43 ± 12	-	-	3052 ± 143

**Table 9 materials-13-02829-t009:** Results of EIS test.

No.		E_opcr_, [mV]	R_s_, [Ωcm^2^]	R_pore_, [Ωcm^2^]	CPE_pore_, [mV]		R_ct_, [kΩcm^2^]	CPE_dl_, [mV]	
					Y, [kΩcm^−m^s^−n^]	n_1_		Y, [Ωcm^−m^s^−n^]	n_2_
1	M1	−257	40	40	0.8553 × 10^−5^	0.9	4930	0.6732 × 10^−5^	0.7
2	M2	−257	55	619	0.7440 × 10^−6^	0.9	4520	0.7616 × 10^−6^	0.9
3	M3	−257	54	592	0.7440 × 10^−6^	0.6	10,255	0.1884 × 10^−5^	0.9
4	M4	+42	69	1	0.1258× 10^−4^	0.8	2494	0.1759 × 10^−4^	0.8
5	M5	−2	68	53	0.4426 × 10^−4^	0.9	11,320	0.3157 × 10^−4^	0.7

**Table 10 materials-13-02829-t010:** The results of the wear test.

No.	Name	µ	Wear Volume, [µm]	W
1	M1	0.54 ± 0.06	51,571 ± 9184	2.2 × 10^−16^
2	M2	0.55 ± 0.08	19,174 ± 1131	7.7 × 10^−17^
3	M3	0.62 ± 0.6	22,287 ± 1063	8.9 × 10^−17^
4	M4	0.63 ± 0.07	19,247 ± 1053	7.4 × 10^−17^
5	M5	0.69 ± 0.07	18,786 ± 1088	7.2 × 10^−17^

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
