# Peer review of "The Influence of Hybrid Surface Modification on the Selected Properties of CP Titanium Grade II Manufactured by Selective Laser Melting"

_materials, 2020, doi:10.3390/ma13122829_

Round 1

Reviewer 1 Report

  1. In section 3.2, the author has discussed the texture and coating effects on the wetting angle and surface free energy of distilled water. However, there is no discussion about the results of the sample that dripped by diiodomethane Please add the discussion about the results of diiodomethane drops.
  2. In Table 7, it shows a different tendency of wetting angle between the distilled water and diiodomethane. Why?
  3. In Table 7, it shows a tendency of wetting angle decrease of the samples with textured surface. There is only a sentence explaining this phenomenon in discussion. Please add more description and related references about this phenomenon.

Author Response

13.06.2020

Professor Marcin Adamiak

Silesian University of Technology

Konarskiego 18A st. 44-100 Gliwice

Dear Reviewer,

Thank you very much for the amendments to our article entitled “The influence of hybrid surface modification on the selected properties of CP titanium Grade II manufactured by selective laser melting”. We appreciate the work you've done to make our article better. We've made the necessary adjustments, taking into account the comments. we wish to send out a revised version and respond to the allegations made:

  • In section 3.2, the author has discussed the texture and coating effects on the wetting angle and surface free energy of distilled water. However, there is no discussion about the results of the sample that dripped by diiodomethane Please add the discussion about the results of diiodomethane drops.
  • In Table 7, it shows a different tendency of wetting angle between the distilled water and diiodomethane. Why?

The measure of the wetting angle using diiodonomethane as a measure liquid are performed in order to the surface free energy calculation. Measurement with diiodomethane as the reference liquid does not provide information on the chemical nature of the surface, i.e. the degree of hydrophilicity / hydrophobicity. Surface properties, including wettability and charge, influence the absorption of protein, which mediate the adhesion of bacteria and cells to the implant. The adsorption of proteins is time depend phenomena, which happen in steps. First, when the material is inserted in the biological environmental, the water molecules adsorption to its surface is happens. The reason is simply the abundance of water in the body. Many molecules, including proteins are dissolved in the aqueous medium. Therefore, understanding the wetting behaviour of the material surface with water becomes an essential step, which has been focused in the present work. The different tendency of wetting angle between the distilled water and the diiodonomethane are associated with their different chemical character – the water is a polar and diiodonomethane is apolar liquid, which differ in their vapour pressure at room temperature and the molecular volume. On the weakly polar surface of the material, the lower values of the contact angles were measured for diiodomethane droplets – for all tested samples group the value of the contact angles measured using diiodomethane were similar and were in the range 44-51 ° and higher compared to polar components. The explanations are included in the "discussion" section. In Table 7, it shows a tendency of wetting angle decrease of the samples with textured surface. There is only a sentence explaining this phenomenon in discussion. Please add more description and related references about this phenomenon.

The value of the contact angle for the samples after laser texturing (M4 and M5) decreased, which can be associated with the development of surface topography and chemical heterogeneity, thus inducing deviations from an ideal surface. Due to increased surface area, there was an increased wettability of the textured surface against deionized water. After laser texturing, in the effect of the condensation of the evaporated material, some buildups around the scanning path were formed. Based on the obtained results, it can be concluded that the formed frames cannot support the droplet of the water, and the droplet immersed into the textured, which is associated with the Wenzel state. However, the values of the wetting angle for both M4 and M5 are approximately 80 °, which indicates a slight advantage of gravitational forces over surface tension forces. The laser scanning speed affected the height and width of the frame, and the side length of the hexagonal honeycomb affected the value of. When the height of the convex hexagonal frame was suitable, the liquid is difficult to enter the groove on the surface, so the droplet was held up by the raised frames and air pockets were formed between the water drop and rough surface, which exhibited Cassie-Baxter state. In the future, research is planned into the effect of texturizing optimization on surface wettability.

Additionally, all suggestions for improving the figures were included.

We hope that the adjustments made to meet your expectations. We wish you a lot of health and patience in these difficult times.

Sincerely,

Reviewer 2 Report

General comments:

In principle, the paper is too long and contains lots of information that is not useful for this paper. It should be significantly shortened and restructured to make it more “reader – friendly”. For instance, explanations about layer deposition techniques are far too detailed. These things can be found in any book, but they should not be a part of scientific paper.

Significant changes in introduction and discussion part are necessary. In the introduction, there is too much irrelevant information, not directly related to the presented research. In particular, there is too much about biocompatibility and reaction of human body. There are no measurements of biocompatibility in the paper, so this part is not needed here and only one sentence would be enough. Introduction should, however, introduce better the presented work. There should be described motivation and information (with references to literature in this topic) about: the choice of the used materials (there is only one sentence and this could be elaborated), why particularly this type of surface modification is studied and why particularly these properties, which are studied here are important. I am missing in the paper information what do we learn from polarization and EIS measurements and why are they important. The motivation for these measurements (why such parameters play a role for biocompatibility) should be described in introduction.

On the other hand, discussion contains a lot of information that belongs to the introduction, while it should be focused only on the results presented in the “results” section. There is very little in the discussion about performed electrochemical measurements. This part is clearly missing here.

Minor coments:

54: “A high friction coefficient and low wear resistance favors in degradation of materials my lead to release and accumulation of elements / corrosion products which lead to a disease called metallosis.” - sth wrong with this sentence, please rewrite

91: Was the SEM image in Figure 1 taken by the authors (then some comments in experimental part are needed as SEM is mentioned only in the context of printed parts) or is it provided by the powder producer (then a reference is needed).

122, 126: Figure 2 has wrong number (1), but this figure is badly described (what are 4- elements?, where is 5 in the fig.?) and the difference between a and b is not understandable. Actually this figure is not needed in the article at all, so it should be removed. It does not bring any new information and there is no need to explain such standard methods in a scientific publication.  

163: the authors should call it optical microscope, not digital. In the end all the recorded images from optical and electron microscopes are digital, but the technique of generating and detecting signal is important for the analysis.

176 I think the authors mean “PICcel3D detector”, which is a 2D detector, and not “PICcel 3D detector”, which would mean that it is a 3D detector. I suppose PICcel3D is a 2D detector that can be used for generating 3D data, but please clarify it.

236: “structure was homogenous and no unmelted powder particles and porosity or other defects could be seen” – there is no evidence for it, or? Based on what do authors claim that there is no porosity?

257: “It can be seen that the samples with CrN layers are silver-gray in color, while the samples with TiN layer exhibit a golden appearance.” – is this information useful ? if yes, pleas elaborate why is this relevant, if not, remove it.

260: the caption of the fig. 4 needs to be improved (include more information e.g. EDS, XRD of what?)

The EDS analysis do not bring any information. Authors show here that samples made of Ti contain Ti and samples made of Ti with a CrN coating contain Ti and Cr…. This EDS data should be removed, as it is useless.

XRD patterns are measured in 2Th range from 20 to 90 deg, but in the “methods” part it is written: “2Θ Bragg angle ranging from 40 to 90°”.  Pattern in c) has not identified peaks – what phase do they correspond to?

280: Fig. 6 d:  it should be written that it is optical microscopy (both in caption and in text before)

286: “Based on the measurements of the depth of  generated textures, no thicknesses of previously refined layers were exceeded – thickness of CrN  layer ~ 2 μm and TiN ~ 4-5 μm.”- this statement is not correct if the thickness is 4.5 μm (as it is written in that paragraph)

339: “obtain”

355: Figures 12-16 should be combined in one figure (as a,b,c,d ….) . There is also a problem with captions (which one is Bode and which one is Nyquist diagram)

360: “The analysis of the impedance spectra of corrosive systems tested samples with different surface  condition - Ringer’s solution, the equivalent electrical system shown in Figure 17 was used.” – This sentence is not clear, please rewrite it.

383: “Only for the samples with CrN coating M2 the pitting corrosion was observed (Figure 18).”  How is that possible that the corrosion was observed for M2, but not for M4 (also with CrN) ?

395: “…group, there is a running in period at the beginning of testing, after which…” – This sentence is not understandable (what is running?).

415: “The interaction of surface modification by thin layer deposition in combination with laser  surface textures was widely studied due the good impact of the hybrid surface modification on the  properties such as wear friction or corrosion resistance of base material, which can be used for example for medical applications [45-48]. Additionally, the Selective Laser Melting SLM process is increasingly being used to manufacture biomedical products, what is associated with possible to fabricate very complex geometries and assemblies with high dimensional accuracy, such as modern bone implant systems or cardio stents [26,28,30].”

This whole paragraph belong to introduction and not discussion. Besides:

“interaction of surface modification by thin layer deposition in combination with laser surface textures….”- interaction with what? Maybe authors mean “effect of…” instead of interaction?

423: “humongous”

426: “α-Ti solid solution” ?? there is no such thing as solid solution of one element

What is meant by substrate here? Typically substrate in SLM would be the same thing as build plate, but here I suppose authors meant the printed sample, or?

445: “significant”

Author Response

13.05.06.2020

Professor Marcin Adamiak

Silesian University of Technology

Konarskiego 18A st. 44-100 Gliwice

Dear Reviewer,

Thank you very much for the amendments to our article entitled “The influence of hybrid surface modification on the selected properties of CP titanium Grade II manufactured by selective laser melting”. We appreciate the work you've done to make our article better. We've made the necessary adjustments, taking into account the comments. we wish to send out a revised version and respond to the allegations made:

General comments:

In principle, the paper is too long and contains lots of information that is not useful for this paper. It should be significantly shortened and restructured to make it more “reader – friendly”. For instance, explanations about layer deposition techniques are far too detailed. These things can be found in any book, but they should not be a part of scientific paper.

Significant changes in introduction and discussion part are necessary. In the introduction, there is too much irrelevant information, not directly related to the presented research. In particular, there is too much about biocompatibility and reaction of human body. There are no measurements of biocompatibility in the paper, so this part is not needed here and only one sentence would be enough. Introduction should, however, introduce better the presented work. There should be described motivation and information (with references to literature in this topic) about: the choice of the used materials (there is only one sentence and this could be elaborated), why particularly this type of surface modification is studied and why particularly these properties, which are studied here are important. I am missing in the paper information what do we learn from polarization and EIS measurements and why are they important. The motivation for these measurements (why such parameters play a role for biocompatibility) should be described in introduction.

On the other hand, discussion contains a lot of information that belongs to the introduction, while it should be focused only on the results presented in the “results” section. There is very little in the discussion about performed electrochemical measurements. This part is clearly missing here

Yes, we agree that the first version was too chaotic and contained too much unnecessary information. The Introduction has been reorganized. An explanation of the basic issues was included - why Ti was chosen for the research, we motivated the choice of the chemical method of surface modification, and the purposefulness of the conducted research. The irrelevant information form the discussion has been deleted. The analyses of the electrochemical properties of tested samples were extended.

Minor comments:

  • 91: Was the SEM image in Figure 1 taken by the authors (then some comments in experimental part are needed as SEM is mentioned only in the context of printed parts) or is it provided by the powder producer (then a reference is needed).

The SEM image of the powder material was taken by the authors, using Scanning Electron Microscope Supra 35 (Zeiss). The image has been standardized relative to the others, which were presented in work.

  • 176 I think the authors mean “PICcel3D detector”, which is a 2D detector, and not “PICcel 3D detector”, which would mean that it is a 3D detector. I suppose PICcel3D is a 2D detector that can be used for generating 3D data, but please clarify it.

We used PIXcel 3D detector.

  • 236: “structure was homogenous and no unmelted powder particles and porosity or other defects could be seen” – there is no evidence for it, or? Based on what do authors claim that there is no porosity?

The surface of the samples subjected to surface modification was examined using microscopic observation. The surface was free from defects. The statement was intended only to determine the state of the surface preparation before coatings deposition. This does not mean, that porosity did not occur at the samples material. The density of the obtained samples is 4.31 g / cm3, which is 95% of the reference density.

  • XRD patterns are measured in 2Th range from 20 to 90 deg, but in the “methods” part it is written: “2Θ Bragg angle ranging from 40 to 90°”. Pattern in c) has not identified peaks – what phase do they correspond to?

The XRD patterns are measured in 2Theta range from 20 to 90 degree. Unidentified peaks for the base material correspond to α-Ti phase.

  • 286: “Based on the measurements of the depth of generated textures, no thicknesses of previously refined layers were exceeded – thickness of CrN  layer ~ 2 μm and TiN ~ 4-5 μm.”- this statement is not correct if the thickness is 4.5 μm (as it is written in that paragraph).

It is possible to break the layer as a result of texturing. The statement has been corrected

  • 383: “Only for the samples with CrN coating M2 the pitting corrosion was observed (Figure 18).” How is that possible that the corrosion was observed for M2, but not for M4 (also with CrN) ?

The laser texturing leads to improvement of corrosion resistance and it is possibly due to the presence of oxide. The oxide layers, formed as a result of laser texturing in contact with corrosive environments, were characterized by greater stability, and provides better protection against surface digestion.

-426: “α-Ti solid solution” ?? there is no such thing as solid solution of one element

What is meant by substrate here? Typically substrate in SLM would be the same thing as build plate, but here I suppose authors meant the printed sample, or?

The word base was used in the context of the basic material.

Additionally, all suggestions for improving the figures were included.

We hope that the adjustments made meet your expectations. We wish you a lot of health and patience in these difficult times.

Sincerely

Reviewer 3 Report

This manuscript describes the coating of c.p. Ti by SLM for biomedical applications. There are some issues whch are not clear.

Why c.p. Ti grade II? What about Ti grade IV or Ti alloy as Ti6Al4V?

Why this materials for coating? Is it possible other combinations?

Why these thicknesses of the coatings? What is the effect of the thickness?

Lines 62-65: it is well-known that surface modification improves biocompatibility. But, PVD methods are not the only technique. Other methodology to modifiy the surface should be mentioned.

Line 86-88: the style of the paragraph is different.

Table 2: why density is in a different column? It should be reordered

Table 4: details about grinding and polishing are not indicated in the work

Figure 3b: units are missing

Figure 4: EDS and XRD should be normalized, so results could be compared.

Figure 7b and 8b: there is not scale bar and the indication of the depth measurement location is not visible.

Figure 9: there is not indication. It is difficult to see if the angle are larger or not.

Figure 10-11: I think if should be joint to be compared

Figure 19b: units of high are missing

Line 444-446: "Based on the studied provided by Albrektoss et al. [53] it can be concluded that the surface finish and surface properties of biomaterials have signification effect on the biological response to an implant, especially healing response and formation of new bone around the implant"

However, it is not checked in these samples. How are the authors so sure about the results?

Please check the style, according to the Journal:

  • Line 86-88: the style of the paragraph is different.
  • Parameters are in italic
  • Name of the company and country of the equipment is not especify in all of them.
  • Exponential are not written in the same way along the manuscript

Author Response

13.06.2020

Professor Marcin Adamiak

Silesian University of Technology

Konarskiego 18A st. 44-100 Gliwice

Dear Review,

Thank you very much for the amendments to our article entitled “The influence of hybrid surface modification on the selected properties of CP titanium Grade II manufactured by selective laser melting”. We appreciate the work you've done to make our article better. We've made the necessary adjustments, taking into account the comments. we wish to send out a revised version and respond to the allegations made:

  • Why c.p. Ti grade II? What about Ti grade IV or Ti alloy as Ti6Al4V?

We would like to show research carried out to the widest spectrum of possibilities as possible. The Titanium is an element that is included as a core element in almost every medical application. We would like to show the influence of the surface modification on titanium itself without any alloying elements that may influence the results. The pure titanium is one of the most popular materials for dental application and we would be focused on the physicochemical and electrochemical properties. In an aspect of the hybrid surface modification and their impact on the physicochemical properties, the results obtained for the pure titanium and for the Ti-6Al-4V alloy may be similar. For both material the most commonly identified layer is TiO2, showing two different crystalline phases, anatase and rutile.

In addition, as part of the work carried out for dental applications. Due to the fact that we use test samples, made with SLM technology, the printing process using GRADE II has been mastered by us at an acceptable level.

  • Why this materials for coating? Is it possible other combinations?

Yes, there are numerous different combinations. We have made the justification in the text. We were guided by the application potential and ease of execution. The TiN and CrN coatings, which are designed to increase not only the corrosion resistance of pure titanium but also to changing the wetting angle. Besides, these coatings improve tribological properties, which is important due to the poor resistance to abrasion of titanium-based materials.

  • Why these thicknesses of the coatings? What is the effect of the thickness?

The thickness is closely related to the desire to obtain the appropriate roughness. A situation in which the coating layer is punctured into the base material after laser treatment is not acceptable. Coating thickness measurements were carried out to characterize them. The choice of coating deposition method and type of coating are often factors determining its thickness (to obtain a compromise between internal stress and thickness).

  • Lines 62-65: it is well-known that surface modification improves biocompatibility. But, PVD methods are not the only technique. Other methodology to modifiy the surface should be mentioned.

We made corrections to that fragment. They are significant, but our scope is to explain and demonstrate the utility of the methods used in this article. No less according to your remarks we mentioned other methods of surface modification, I hope not too cursory.

Currently, it is possible to use more surface modification of metal materials using for medical applications. There are numerous different combinations. Generally, surface modification can be divided into groups: physical and chemical methods. The physical methods lead to change in the topography or morphology, for example, etching, grit-blasting. However, physical methods result also in little to no change in chemistry. However, it was decided to focus mainly on the impact of surface modification on physicochemical properties. Well-established chemical methods of surface modification include plasma and chemical vapor deposition, atomic layer deposition, and electrochemical depositions. We were guided by the application potential and ease of execution. The choice of surface layer deposition technology often limits the choice of coating type. However, among the many different types of coatings currently used for biomedical purposes, very common and relatively easy to perform are TiN and CrN coatings, which are designed to increase not only the corrosion resistance of pure titanium but also to changing the wetting angle. The chemistry character of the surface has a significant effect on the value of the wetting angle and surface free energy, which are an important factor that could affect cellular response and bacterial plaque accumulation. In addition, laser texturing was performed on the surface topography. However, as a result of physical phenomena occurring during the interaction of the laser beam with the substrate material, there is also a change in the chemical nature of the surface. As a result, the formation of oxides on the surface of the material increases.

  • Table 4: details about grinding and polishing are not indicated in the work

The details about grinding and polishing has been described extensively.

  • Figure 4: EDS and XRD should be normalized, so results could be compared.

EDS analysis results have been removed as suggested by another reviewer. In the case of XRD analysis - a conventional scale with a.u. was introduced. It is not possible to normalize the scale - the peak for the TiN layer reaches 70,000 cps, in the case of base material the highest peak corresponds to 800 cps.

  • Line 444-446: "Based on the studied provided by Albrektoss et al. [53] it can be concluded that the surface finish and surface properties of biomaterials have signification effect on the biological response to an implant, especially healing response and formation of new bone around the implant"

However, it is not checked in these samples. How are the authors so sure about the results?

We believe that the argument used is overinterpretation. The aspects of biological response to the surface finishing of the implants were not included in the manuscript – no microbiological tests were performed. Studies, which were performed may indicate that the chemistry character of the surface and surface topography of the biomaterials may have a significant effect on the biological response to an implant. However, in the case of surface modification presented in this work, it requires confirmation in the future studies. Further research is required. In the next stage of research, biological studies are planned. Paragraph has been deleted

Additionally, all suggestions for improving the figures were included.

We hope that the adjustments made meet your expectations. We wish you a lot of health and patience in these difficult times.

Sincerely,

Round 2

Reviewer 2 Report

The authors clearly put a lot of effort to improve the paper. Now the article is more compact and therefore it reads well.

In my opinion the article should be published after minor corrections:

  1. I see I was not clear about the X-ray detector : the name is PIXcel3D and not "PiXcel 3D" detector (when it is written with a space it means that it is a 3D detector). So what I mean, is that the space should be deleted. So far there are available 1D and 2D X-ray detectors, but no 3D detectors.  Yes, it is a minor detail, but...
  2. I am not convinced about the statement of the authors that "x" peaks in XRD patterns correspond to aplha Ti. They are clearly in different positions than in the pattern in c) (the one for pure Ti).
  3. The strongest peak for CrN marked in the pattern is now (222), but in text it is written (113) (so which one is correct?). If the authors indicate lattice planes related to diffraction peaks, they should also indicate which space group are they reffering to (for CrN it is not so obvious as for aplha Ti)
  4. I see the authors decided to keep the word "humongous", which is rather colloquial language and it means "huge". Is it really the word that authors intended to use? If yes, I don't understand this sentence.

Author Response

19.05.06.2020

Professor Marcin Adamiak

Silesian University of Technology

Konarskiego 18A st. 44-100 Gliwice

Dear Reviewer,

Thank you very much for the next amendments to our article entitled “The influence of hybrid surface modification on the selected properties of CP titanium Grade II manufactured by selective laser melting”. We appreciate the work you've done to make our article better. We've made the necessary adjustments, taking into account the comments. we wish to send out a revised version and respond to the allegations made:

  • I see I was not clear about the X-ray detector : the name is PIXcel3D and not "PiXcel 3D" detector (when it is written with a space it means that it is a 3D detector). So what I mean, is that the space should be deleted. So far there are available 1D and 2D X-ray detectors, but no 3D detectors.  Yes, it is a minor detail, but...

The name has been corrected. Of course, PIXcel3D the  is a unique 2D solid-state hybrid pixel detector.

  • I am not convinced about the statement of the authors that "x" peaks in XRD patterns correspond to alpha Ti. They are clearly in different positions than in the pattern in c) (the one for pure Ti).
  • The strongest peak for CrN marked in the pattern is now (222), but in text it is written (113) (so which one is correct?). If the authors indicate lattice planes related to diffraction peaks, they should also indicate which space group are they referring to (for CrN it is not so obvious as for alpha Ti)

In fact, the recorded XRD patterns presented the peaks, which have not been clearly identified – thank you for paying attention. The strongest peak for CrN coating in the XRD pattern is (113) – the peak on the diffraction pattern  was incorrectly indicated as (222). The peaks marked “x” most likely correspond to Cr, which indicates that the obtained coating is not stoichiometric. For this reason, the strongest peak for CrN coating occurs for the CrN (113) plane and not for the CrN (002) as in the case of stoichiometric coatings.

In the case of TiN coating, only one of the recorded peaks does not correspond to the alpha Ti phase. Most likely, the peak may correspond to Ti oxide (rutile), but the peak intensity is low and no other peaks in this phase were recorded, which makes identification difficult.

In addition, the XRD analysis was performed to complement the EDS analysis, which was to confirm the presence of TiN and CrN coatings on the base material (Grade II). For this reason the XRD analysis has not been the subject of the in-depth reporting. Authors are aware that in the case of coatings obtained in PVD processes, many factors, such as stoichiometry of the obtained phases, internal stresses, may affect the intensity of the recorded diffractograms and affect the shifts of the recorded reflections, however, no such analyzes were performed in these studies.

  • I see the authors decided to keep the word "humongous", which is rather colloquial language and it means "huge". Is it really the word that authors intended to use? If yes, I don't understand this sentence.

The word “humongous” has been replaced by the correct term “homogeneous”.

We hope that the adjustments made meet your expectations. We wish you a lot of health and patience in these difficult times.

Sincerely

Reviewer 3 Report

Dear authors

Thank you for introducing the proposed modifications.

Please be aware to be uniform. Grade 2 and grade II are written.

Also, no decimals are included in scale bar of the images

Author Response

19.06.2020

Professor Marcin Adamiak

Silesian University of Technology

Konarskiego 18A st. 44-100 Gliwice

Dear Reviewer,

Thank you very much for the next amendments to our article entitled “The influence of hybrid surface modification on the selected properties of CP titanium Grade II manufactured by selective laser melting”. We appreciate the work you've done to make our article better. We've made the necessary adjustments, taking into account the comments. we wish to send out a revised version and respond to the allegations made:

  • Please be aware to be uniform. Grade 2 and grade II are written.

The text has been unified.

  • Also, no decimals are included in scale bar of the images

We suppose that the remark refers to the figure 12. The decimals have been included in results of the width and height measurements.

We hope that the adjustments made meet your expectations. We wish you a lot of health and patience in these difficult times.

Sincerely